# A Review of Recent Developments in Smart Textiles Based on Perovskite Materials

**Madeeha Tabassum [1], Qasim Zia [2], Yongfeng Zhou [1], Yufei Wang [1], Michael J. Reece [1] and Lei Su [1,\*]**

[1]  School of Engineering and Materials Science, Queen Mary, University of London, London E14NS, UK
[2]  Department of Materials, The University of Manchester, Oxford Rd, Manchester M13 9PL, UK
[\*]  Correspondence: l.su@qmul.ac.uk

**Abstract:** Metal halide perovskites (MHPs) are thought to be among the most promising materials for smart electronic textiles because of their unique optical and electrical characteristics. Recently, wearable perovskite devices have been developed that combine the excellent properties of perovskite with those of textiles, such as flexibility, light weight, and facile processability. In this review, advancements in wearable perovskite devices (e.g., solar cells, photodetectors, and light-emitting diodes) concerning their device architectures, working mechanisms, and fabrication techniques have been discussed. This study also highlights the technical benefits of integrating MHPs into wearable devices. Moreover, the application challenges faced by wearable perovskite optoelectronic devices—from single devices to roll-to-roll manufacturing, stability and storage, and biosafety—are briefly discussed. Finally, future perspectives on using perovskites for other wearable optoelectronic devices are stated.

**Keywords:** perovskite; flexible substrate; textile; optoelectronics

## 1. Introduction

Wearable and flexible optoelectronics have been the subject of many studies due to their promising applications in various emerging fields, including healthcare systems, flexible displays, sensors, and human activity monitoring. The dominant features of wearable optoelectronic devices are flexibility, light weight, and large-area processability with low-temperature flexible substrates [1,2].

Metal halide perovskites (MHPs) have received considerable attention in textile-based flexible optoelectronic devices due to their high optical absorption coefficient, efficient charge carrier mobility, high photoluminescence quantum yield (PLQY), and low-cost solution fabrication techniques. Perovskites have an agreed stoichiometric formula of $ABX_3$, where A is a monovalent cation (either organic (e.g., $CH_3NH_3^+$ (methylammonium/$MA^+$), $CH(NH_2)_2^+$, (formamidinium/$FA^+$)) or inorganic (e.g., $Cs^+$ (caesium)), B is a divalent cation (mostly $Pb^{2+}$ (lead) or $Bi^{3+}$ (bismuth)), and X is a halogen anion ($Cl^-$, $Br^-$, or $I^-$) [3–6]. MHPs are considered important materials by a growing part of the research community due to their application in solar cells, light-emitting diodes (LEDs), photodetectors (PDs), lasers, and nonlinear optics [7].

In 2009, after Kojima et al. demonstrated the first perovskite device, MHPs have been the subject of great interest in a wide range of optoelectronic devices [8]. Kumar et al. reported the first flexible solar cells, with a power conversion efficiency (PCE) of 2.6% [9]. Recent advances in flexible perovskite devices and understanding of MHPs have facilitated a boost in the PCE of solar cells by more than 20% [10]. Similarly, the first flexible LEDs were fabricated by Bade et al. in 2016. In this study, printed metal halide perovskite LEDs were reported, where carbon nanotubes (CNTs) and silver nanowires were used as the anode and cathode, respectively. A composite film made of $MAPbBr_3$ and polyethylene oxide was used as an emissive layer in the LEDs. The devices on CNTs/polymer were able to handle a radius curvature of 5 mm, and had an external quantum efficiency (EQE)

of 0.14% [11]. To improve the efficiency of flexible LEDs, the perovskite emissive layer has been constantly improved, and the EQE has exceeded 20% [12]. Moreover, the first MHP-based broadband photodetector was reported on a flexible ITO-coated substrate. The flexible perovskite photodetector was sensitive to a broadband wavelength from the ultraviolet to the entire visible region, with measured photoresponsivity of 3.49 AW$^{-1}$ and 0.0367 AW$^{-1}$ at 365 nm and 780 nm wavelengths, respectively [13].

As solution-processable MHP offers opportunities for the fabrication of wearable, lightweight, portable, and bendable optoelectronic devices; perovskite-based wearable devices have become the focus of a wide range of electronic devices [14,15]. Textile-based flexible perovskite devices have been developed by various methods, such as vapour deposition [16], inkjet printing [17,18], and other roll-to-roll printing methods [19].

Recently, considerable literature has emerged around the theme of wearable opto-electronics and their application in intelligent devices, such as flexible solar panels and intelligent sensors. For textile-based wearable optoelectronics, mechanical flexibility is an important parameter. However, conventional perovskite-based optoelectronic devices are fabricated on rigid and brittle substrates, hindering their use in wearable electronics. One of the greatest challenges is the special weight and layout requirements for wear-able optoelectronics. Optoelectronic devices synthesised on rigid substrates are not the proper choice for smart wearables. Thus far, flexible optoelectronic devices have been fabri-cated on substrates such as polyethylene terephthalate (PET) or polyester, poly (ethylene 2,6-naphthalate) (PEN), and polyimide (PI).

Indium tin oxide (ITO) is the most widely used electrode material due to its excellent optical transparency and low electrical resistance, although its rigidness limits its use in flexible devices [20,21]. Although many plastic-based flexible and bendable substrates, such as PET, have been reported to replace the rigid electrode [22–24], the has authors have concerns about their ability to be wearable, breathable, and conformable [25]. In addi-tion, electron-transport layers (ETLs) and hole-transport layers (HTLs) are also important components of textile-based flexible perovskite devices. Their optical transparency and high charge conductivity are the main factors to evaluate the performance of wearable perovskite optoelectronic devices [26].

The purpose of this paper is to review recent research and development into textile-based flexible perovskite devices. The specific objectives of this study are to explore the device architectures, working mechanisms, fabrication techniques, and recent advances in fibre- and fabric-based perovskite optoelectronic devices. This study seeks to perform a comprehensive comparison of available data, which will help to address the research gaps in textile-based perovskite devices. Moreover, the challenges of wearable perovskite devices are proposed, and future directions are discussed.

## 2. Perovskite Device Structures and Working Mechanisms

### 2.1. Solar Cells

The device configuration is a key aspect of analysing the overall efficiency of perovskite solar cells. Flexible PSCs have a similar configuration to that of rigid devices. PSCs are mostly categorised as regular (n-i-p) and inverted (p-i-n) architectures, depending on which carrier transport layer (ETL/HTL) is encountered by the solar light first [27–31]. These two architectures can be further classified into mesoscopic and planar. A mesoporous layer is integrated into the mesoscopic structure, while all of the planar layers are used in the planar structure. Perovskite solar cells without electron- and hole-transport layers have also been reported. In brief, six types of PSCs have been studied by various researchers: n-i-p structure (planar), p-i-n structure (planar), n-i-p structure (mesoscopic), p-i-n structure (mesoscopic) (Figure 1a–d), and ETL- and HTL-free structures. In a typical device structure, perovskites act as a photoactive layer and absorb the light to generate the free charges. Electrons and holes are transferred by ETLs and HTLs to the cathode or anode to avoid the chance of charge recombination [26,32,33].

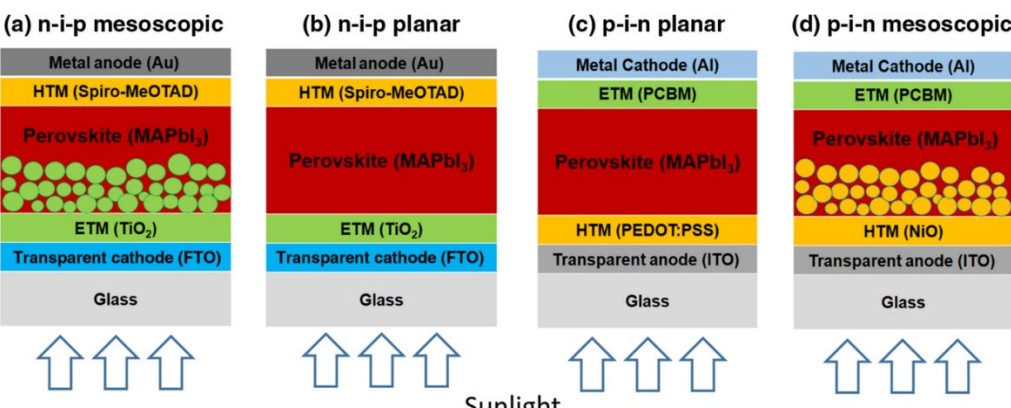

**Figure 1.** Schematic diagram showing the four typical layered structures of perovskite solar cells: (**a**) n-i-p mesoscopic; (**b**) n-i-p planar; (**c**) p-i-n planar; (**d**) p-i-n mesoscopic. Reproduced from [33], copyright 2018, Springer Nature.

### 2.2. Light-Emitting Diodes (LEDs)

Typical perovskite LEDs adopt one of two configurations: conventional or inverted, which mostly refer to n-i-p or p-i-n structures, respectively. For the conventional layout, LEDs are generally composed of a glass substrate covered with a transparent conductive electrode such as ITO to work as a cathode. This electrode injects an electron into the ETL and then into the perovskite emissive layer (EML). Similarly, the holes are injected from the HTL to the perovskite layer for radiative recombination with electrons. For an inverted layout, the electrons and holes are injected in the opposite direction, where ITO works as an anode to inject holes into the HTL and then into the perovskite emitter. At the same time, electrons are injected from the cathode to the ETL to enter the EML, where radiative recombination occurs [34].

Ideally, the device structure is designed in such a way that electrons and holes injected from the electrodes recombine radiatively in the perovskite EML, giving rise to the emission of photons [35,36]. For the fabrication of textile-based perovskite LEDs, flexible substrates and the choice of electrodes are the key factors. These are important components that define the mechanical behaviour, surface roughness, breathability, and wearability of textile-based perovskite LEDs [11].

### 2.3. Photodetectors

MHPs have emerged as the most efficient and low-cost energy materials for diverse optoelectronic and photonic device applications. Recently, many exciting results on perovskite-based light detector devices have been reported. Different device configurations of perovskite photodetectors have been developed, and demonstrated a significant increase in photodetection efficiency. MHP-based photodetectors are generally divided into photodiodes, photoconductors, and phototransistors. Photodiodes, also known as vertical-structure photodetectors, consist of photoactive material sandwiched between two electrodes. The vertical configuration of the photodetectors is structured from planar heteronode PSCs; hence, this layout can be roughly divided into regular (n-i-p) and inverted (p-i-n) categories [2,37]. The lateral-structure photodetectors comprise photoconductors and phototransistors, which can detect light and signal amplification. Photoconductors have the simplest device structure, which consists of a photoactive material and two ohmic metal contacts forming a metal–semiconductor–metal configuration. Due to the considerable distance between the two electrodes, photogenerated charge carriers take a long time to reach the electrodes; hence, the slow response time and high driving voltage [38].

For textile-based flexible photodetectors, flexible substrates, flexible electrodes, and functional layers are the main elements. To date, a variety of flexible substrates—including carbon cloth, fibre, paper, PET, PI, and polydimethylsiloxane (PDMS)—have been reported in flexible perovskite photodetectors [34].

### 3. Manufacturing Techniques

Due to the chemical nature of MHPs, perovskite thin films can be synthesised by using a precursor solution. The processability of perovskite solutions can be made possible by the lab-scale spin-coating method, which has provided the most efficient PSC devices, with a PCE of over 25% being the best result reported at present [39]. At the commercial scale, solution processing is also compatible with roll-to-roll (R2R) manufacturing [40,41]. R2R production is an ultimate solution to fabricate large-area modules in terms of low cost and high output. Printing/coating techniques that are compatible with R2R manufacturing include inkjet printing, slot-die coating, blade coating, and spray coating [27].

#### 3.1. Spin Coating

Spin coating (Figure 2a) is a convenient and widely used solution-based method to fabricate uniform and pinhole-free perovskite thin films. By using this method, the compact thin films can be directly deposited on a variety of substrates (e.g., glass, quartz, plastic, and silicon) from a precursor solution made of metal halides and organic halides. In addition to spinning speed and time, a post-annealing treatment at low temperature ($T < 25\,^{\circ}C$) is essential to increase the phase purity and crystallinity of perovskites [42,43]. The spin coating can be categorised into one-step and two-step methods for the fabrication of perovskite thin films with thicknesses ranging from 10 nm to 100 nm. Generally, the one-step spin coating involves anti-solvents. Chlorobenzene and toluene are the mostly widely used anti-solvents during the spin coating of MHP in N,N-dimethylformamide (DMF) and a mixed solvent of dimethyl sulfoxide (DMSO) and γ-butyrolactone (GBL) [44]. The precise control of the anti-solvent with respect to time is the main weakness of the one-step spin-coating method for the production of large thin films. On the other hand, the two-step spin-coating method was found to be efficient because of its better morphology and interface control. However, incomplete conversion of Pb-based salts into perovskite is a serious issue related to the two-step spin-coating method, limiting its large-scale production with good repeatability. For both methods, the waste of perovskite-based precursor materials indicates that spin coating is a lab-scale method to fabricate wearable and flexible perovskite-based smart optoelectronic devices with small areas. Another reason to use spin coating at the lab scale is that with the large area, pinholes and non-uniform thickness are more likely to appear in the surface morphology, leading to the loss of final device performance [27,45].

#### 3.2. Thermal Evaporation

Thermal evaporation (Figure 2b) is another lab-scale method to form smooth and uniform perovskite thin films. This method converts perovskite precursor materials into the vapour phase via heating inside the vacuum chamber. The vapour particles produced move towards the substrates where they settle, and a uniform perovskite thin film is deposited. This method can be used to make flexible perovskite optoelectronic devices, as it does not work at high temperatures during the deposition process. However, the production cost for the setup of the vacuum process makes this method complex for the large-scale production of flexible perovskite-based devices [45,46].

There are two main types of this method: single-source and dual-source evaporation. Single-source evaporation uses one precursor material (organic or inorganic) from one source, and the rest of the precursors are used via other production techniques, such as spin coating or blade coating. For dual-source evaporation, both inorganic and organic precursor materials are evaporated at the same time [47]. Era et al. first reported the dual-source evaporation of lead iodide ($PbI_2$) and organic ammonium iodide for the synthesis of layered perovskite thin films [48]. Recently, for the first time, all-vacuum-processed PSCs using an inverted architecture, with PCE of 19.4% for small areas ($0.054\ cm^2$) and 18.1% for large areas ($1\ cm^2$), were reported [49]. Similarly, all-vacuum-processed perovskite thin films were also fabricated for yellow perovskite LEDs. By co-evaporation of CsI and $PbBr_2$, highly smooth and uniform perovskite thin films with a small grain size of ~31.8 nm were

achieved, and demonstrated an EQE of ~3.7% and luminance of ~16,200 cd/m$^2$ [50]. Using the vacuum deposition technique, dual-phase CsPbBr$_3$-CsPb$_2$Br$_5$ perovskite thin films were synthesised for high-performance rigid and flexible PDs. The device had a responsivity and detectivity of 0.375 AW$^{-1}$ and 10$^{11}$ jones, respectively [51]. Therefore, the use of the thermal evaporation technique highlights the great potential to produce highly uniform perovskite thin films in the production of optoelectronic devices.

### 3.3. Roll-to-Roll Manufacturing

3.3.1. Inkjet Printing

Printing methods are mostly recommended for patterned device fabrication, including contact and non-contact printing techniques. The most commonly used non-contact printing method is inkjet printing (Figure 2e). This method uses an array of fine nozzles in a controlled manner to release fine droplets of the precursor solution to generate high-resolution patterns and arrays on different substrates. Several factors—such as printing speed, droplet volume, trajectory, the temperature of the substrate, and environmental conditions—influence the final film characteristics. Inkjet printing has major advantages, including low precursor waste, masklessness, and the ability to fabricate large modules with direct writing technology. The major disadvantage of the inkjet printing method is that perovskite films are discontinuous, with increasing defects, due to rapid ink crystallisation during the printing process. Moreover, unavoidable waste of inkjet material during the deposition process is another problem for the handling of lead-based precursor materials. Many researchers have used this method to fabricate rigid and flexible perovskite optoelectronic devices. Recently, flexible perovskite (CH$_3$NH$_3$PbI$_3$) solar cells were produced based on all-inkjet printing, including the bottom electrode (PEN/Ag), top electrode (Ag nanowires), hole-transport layer (PEDOT: PSS), etc. The results showed a module area of 120 cm$^2$, 150 cm$^2$, and 180 cm$^2$ with PCE of 16.78%, 12.56%, and 10.68%, respectively [18].

In another fully inkjet printing method, flexible and efficient PDs were achieved under ambient conditions. The fully printed perovskite (caesium-doped cation) PDs showed excellent mechanical and electrical stability for 700 h without any encapsulation. PDs exhibited a high detectivity value of 7.7 × 10$^{12}$ jones, I$_{light}$/I$_{dark}$ ratio of 1.83 × 10$^4$, and photoresponsivity value of 1.62 AW$^{-1}$ [52]. It can thus be suggested that inkjet printing can freely print any design pattern with high speed and strong adaptability on various substrates.

3.3.2. Slot-Die Coating

To scale up perovskite optoelectronic devices, fabrication of the perovskite active layer through a scalable printing method is considered as the next move toward industrial bulk production and high output. The slot-die coating method (Figure 2d) includes a moving substrate, slot-die head, and a pumping system for the deposition of perovskite thin films. By carefully adjusting the flow rate, substrate speed, and width of the printing head, highly crystalline perovskite films can be formed [53–55].

Recently, a homemade slot-die coating setup has been reported to prepare fully covered ZnO ETLs, which exhibited good reproducibility and ambient stability for PSCs. With the controlled deposition parameters, a champion device (FTO/ZnO/Cs$_{0.17}$FA$_{0.83}$Pb (I$_{0.83}$Br$_{0.17}$)$_3$/spiro-OMeTAD/carbon) showed a PCE of 10.8% [56]. Another group of researchers formed slot-die-printed tin oxide thin films for hysteresis-free flexible PSCs. The small flexible FPSCs exhibited a high efficiency of 17.18%, and the large flexible module obtained a PCE of over 15% [56].

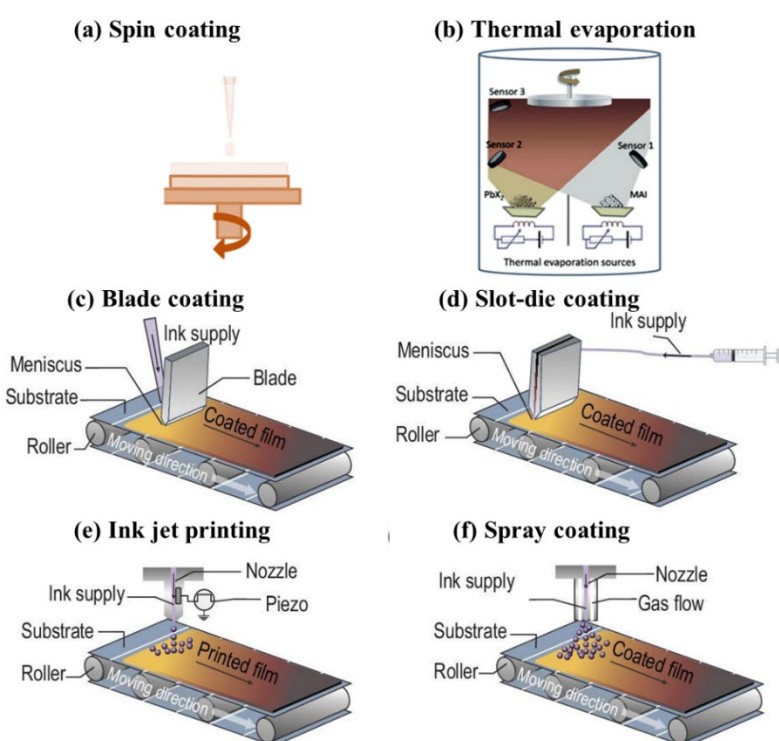

**Figure 2.** Deposition techniques for halide perovskite thin films at laboratory scale and large scale: (**a**) Spin coating; (**b**) thermal evaporation; (**c**) blade coating; (**d**) slot-die coating; (**e**) inkjet printing; (**f**) spray coating. Reproduced from [57], copyright 2021, Oxford University Press.

### 3.3.3. Spray Coating

Spray coating (Figure 2f) is a technique to obtain high-quality thin films over large-area substrates [58]. In this method, the spray coater's tip divides the solution into a fine mist, which is then directed towards the substrate, assisted by an inert gas jet. During the film formation, the spray head passes over the substrate at a fixed height to obtain the required film thickness. The film's morphology closely relies on the head speed, droplet volume, solution concentration, substrate temperature, etc. [59]. This method requires a high temperature (100 °C to 120 °C) to evaporate the precursor solution, which hinders its use on polymer substrates. To date, many researchers have developed low-temperature spray-coating processes to fabricate a variety of perovskite thin films to help make a range of flexible perovskite optoelectronic devices [46].

### 3.3.4. Blade Coating

The blade-coating technique (Figure 2c) has been used for the synthesis of thin films in photovoltaic and optoelectronic devices due to its simplicity, reduced cost, and high deposition rate. The first attempt to replace spin coating for the fabrication of perovskite thin films was the blade-coating technique. During blade coating, a sharp blade is placed in front of the perovskite solution at a suitable position above the target substrate. The linear movement of the blade across the substrate leaves the wet perovskite film [55]. The morphology of the final film is highly dependent on the substrate temperature, so the evaporation rate must be controlled to obtain a good-quality film [58].

$CH_3NH_3PbI_3$ perovskite thin films with large grains and domains at ambient conditions were fabricated for solar cells via the blade-coating method [60]. The film's fabrication process could be optimised by controlling the blade speed, the distance between the blade and the substrate, the solution concentration, and the in situ thermal treatment temperature.

## 4. Textile-Based Perovskite Optoelectronic Applications

*4.1. Wearable Solar Cells*

Textile-based flexible perovskite solar cells have long been a question of great interest because of their unique properties, such as high flexibility, wearability, and ability to conform to any shape. Compared with conventional rigid solar cells, wearable perovskite solar cells can be easily deployed on curved or irregular surfaces of vehicles or tents. The wearability of PSCs mainly depends on the flexibility of the substrates, which defines not only the final efficiency, but also the mechanical and environmental stability. Furthermore, charge-transport layers (HTLs/ETLs) must have better stability against chemicals, oxygen, and water vapour to prevent corrosion and degradation [27,61,62].

The low-temperature synthesis of charge-transport layers and high-quality perovskite films is necessary to produce high-efficiency wearable perovskite solar cells. As mesoporous structure (e.g., $TiO_2$ as the ETL) always demands high-temperature arrangements ($\approx500\ °C$), which are not suitable for flexible substrates, there are only a few scientific reports about the application of this architecture in this field [63–65]. Therefore the advancement in textile-based flexible perovskite solar cells is mostly reported in regular (n-i-p) or inverted (p-i-n) structures [66–68].

The first flexible perovskite solar cell was structured using ZnO nanorods as a mesoscopic scaffold layer and an ETL to allow the fabrication of low-temperature solution-based perovskite $CH_3NH_3PbI_3$ solar cells. A PCE of 8.90% and 2.62% was recorded for rigid fluorine tin oxide (FTO) and flexible PET/ITO substrates, respectively [9]. Sisi et al. reported another effective strategy for the fabrication of textile-based PSCs by synthesising obelisk-like ZnO arrays on stainless steel fabric via a mild solution process. The perovskite $CH_3NH_3PbI_3$ thin layer was formed by the dip-coating process, and the resulting solar cells showed a PCE of 3.3%, with only 7% variation after bending for 200 cycles [69].

A novel stainless steel fibre-shaped PSC with high flexibility and low cost was developed by continuously winding carbon nanotubes on a fibre substrate. Photoactive perovskite $CH_3NH_3PbI_3$ was sandwiched in between them via the solution-processing technique, and the fibre-shaped PSC showed a PCE of 3.3%. The fibre-shaped PSC can be woven into smart textiles for large-scale applications [70].

A recent study to develop a novel fibre-based solar cell textile (Figure 3a) that works at $-40\ °C$ to $160\ °C$ was reported by Limin et al. Briefly, a family of inorganic perovskite solar cell fibres and textiles were made by multiple-sintering techniques to fully cover the curved surface of fibre substrate with large, uniform perovskite crystals. Firstly, $CsPbBr_3$ quantum dots (QDs) were fabricated by a room-temperature ligand-assisted method. Then aligned $TiO_2$ nanotubes were successfully grown on Ti wire and dipped into inorganic perovskite QDs to form a uniform layer on the fibre substrate.

This modified fibre-shaped PSCs were woven into textiles for further applications. The solar cell textile maintained about 90% of its original PCE after bending at $45°$ for 500 cycles, as shown in Figure 3b. The fibre-shaped devices connected in parallel showed that the short-circuit current ($I_{sc}$) increased linearly with the number of devices, whereas the open-circuit voltage ($V_{oc}$) remained unchanged (Figure 3c). For the five devices connected in series, an $I_{sc}$ value of 0.09 mA and a $V_{oc}$ value of 6.22 were achieved (Figure 3d).

After that, a solar cell textile was used to power an electronic watch worn by a human (Figure 3e). In addition, it could also work under harsh working conditions, such as being frozen in ice or placed on red-hot charcoal (Figure 3f,g) [1]. The most recent studies available on textile-based perovskite photovoltaic applications are summarised in Table 1.

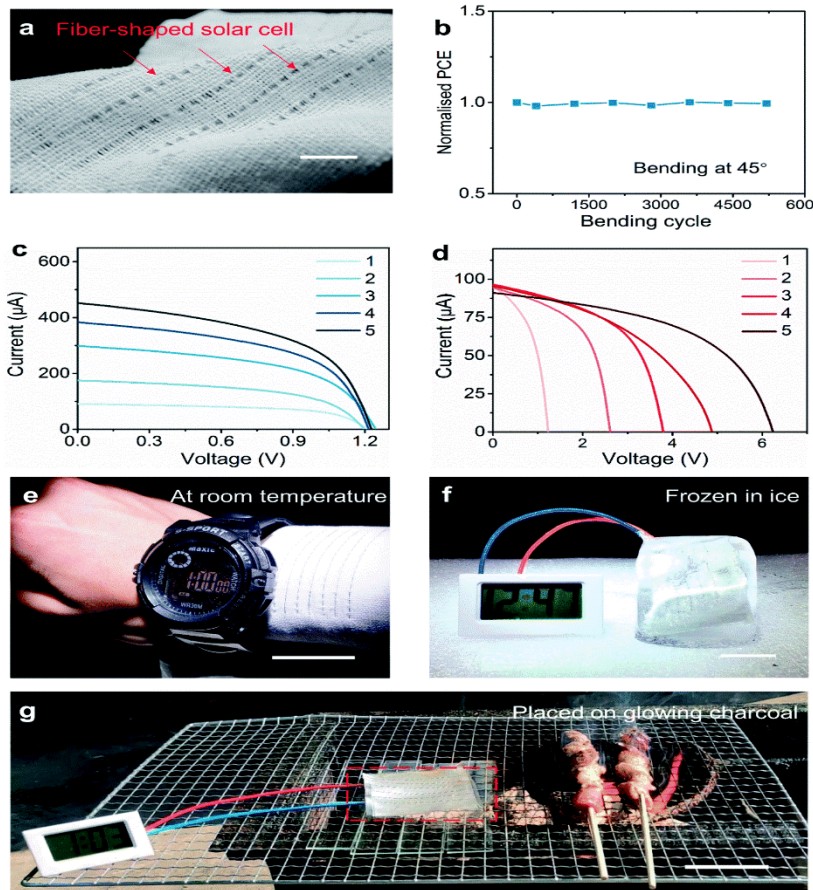

**Figure 3.** Application and performance of the perovskite solar cell textile: (**a**) Photograph of a perovskite solar cell (scale bar = 1 cm). (**b**) Power conversion efficiency with a bending angle of 45° to the solar cell textile. (**c**,**d**) I–V curves of fibre-shaped perovskite solar cells connected in parallel and in series for (**e**) a textile perovskite solar cell powering an electronic watch at room temperature (scale bar = 2 cm), (**f**) a textile perovskite solar cell frozen in ice powering an electronic clock, and (**g**) testing of textile perovskite solar cells placed at charcoal powering an electronic clock (scale bar = 4 cm). Reproduced from [1], copyright 2020, Royal society of chemistry.

**Table 1.** Summary of the textile-based perovskite solar cells' performance [a].

| Textile Substrate | Perovskite Photoactive Layer | Fabrication Method | $V_{OC}$ (V) | $J_{sc}$ (mA/cm$^2$) | FF (%) | PCE (%) | Ref. |
|---|---|---|---|---|---|---|---|
| Polyester fabric | $CH_3NH_3PbI_3$ | Bar coating | 0.88 | 12.44 | 49 | 5.72 | [71] |
| Stainless steel fabric | $CH_3NH_3PbI_3$ | Dip coating | 0.55 | 3.72 | - | 3.80 | [69] |
| Stainless steel fibre | $CH_3NH_3PbI_3$ | Dip coating | 0.66 | 10.20 | 48 | 3.30 | [70] |
| PAN/PU fabric | $CH_3NH_3PbI_3$ | Spin coating | 0.80 | 8.86 | 57 | 4.06 | [72] |
| TiO2 modified Ti-fibre | $CsPbBr_3$ | Dip coating | 1.19 | 6.48 | 70 | 5.37 | [1] |
| Acrylic elastomer | $CH_3NH_3PbI_3$ | Spin coating | 1.06 | 17.05 | 65 | 14.80 | [73] |
| Carbon fabric | $CsMAFAPbI_{3-x}Br_x$ | Spin coating | 1.12 | 20.42 | 67 | 15.29 | [74] |
| PEN | $CH_3NH_3PbI_{3-x}Cl_x$ | Spin coating | 0.96 | 19.06 | 59 | 12.20 | [75] |
| Carbon fibre | $CH_3NH_3PbI_{3-x}Cl_x$ | Dip coating | 0.61 | 8.75 | 56 | 3.03 | [76] |
| Polyester fabric | $(FAPbI_3)_{0.85}(MAPbBr_3)_{0.15}$ | Spin coating | 1.09 | 22.41 | 72 | 17.68 | [76] |

[a] Abbreviations: Ti = titanium; PAN/PU = poly(acrylonitrile)/polyurethane; TiO$_2$ = titanium dioxide.

Jung et al. performed a series of experiments on fully solution-processed perovskite ($CH_3NH_3PbI_3$) solar cells fabricated on PU-coated polyester fabric. A thin layer of PU was coated as a planarisation layer that effectively improved the wettability, processability, and surface morphology of the textile surface. The textile-based flexible PSCs were successfully fabricated, and PCE of 5.72% was achieved by using solution-processed anode, HTL, and ETL materials [71]. In another textile-based PSC report, low-temperature tin oxide

(SnO$_2$) ETL, perovskite (CH$_3$NH$_3$PbI$_3$), and a novel encapsulation layer were obtained. An ITO/PEN flexible substrate was chosen to fabricate the most efficient textile-based PSC with improved wash capability and ambient stability. A 15% PCE of this unique textile-based PSC was recorded, with future potential in wearable device applications [73].

In the development of wearable power sources, highly flexible, lightweight, efficient PSCs based on PEN/ITO substrates with a PCE of 12.2% have been reported. In addition, bending stability was recorded for solar devices with three effective bending radii of 400 mm, 10 mm, and 4 mm for the human neck, wrist, and finger, respectively. In the case of a human finger, the PCE significantly dropped to 50% of the initial value after 1000 cycles. It was noted that the origin of degradation was due to the fracture in the ITO layer on the PEN substrate [75].

### 4.2. Photodetectors for Wearable Optoelectronics

Textile-based photodetectors (PDs) are a major area of interest within the fields of video imaging, bioinspired sensing, optical communication, and biomedical imaging. In recent years, wearable PDs have been fabricated on a variety of flexible substrates because of their possible applications in touchscreens, wearable electronic devices, and pressure-induced sensing [77]. Several key factors define the final efficiency of wearable PDs, such as the morphology of substrates, and the retention of initial performance values after repeated bending, stretching, or folding. Therefore, the main components of wearable PDs—such as substrates, charge-transport layers, and electrodes—should be stable enough to resist environmental and mechanical hazards. In addition, MHPs can be easily synthesised by low-temperature solution-processing techniques, which is helpful in making wearable PDs [78–80].

Dong et al. reported highly flexible fibrous yarn bundles and their knitted structure as a template to fabricate MAPbI$_3$-based PDs (Figure 4e). They fabricated quasi-spring-like network-based wearable PDs consisting of silver (Ag) electrode/perovskite (MAPbI$_3$)/yarn bundles, and their photoelectric properties were examined. In Figure 4a, the I–V curves depict a linear behaviour with an increase in voltage, confirming the ohmic contact between the perovskite and the Ag electrodes. After illumination, the current gradually increased with an increase in the power from 10 mWcm$^{-2}$ to 80 mWcm$^{-2}$, and the time-dependent photocurrent was also recorded under the same power conditions (Figure 4b). During repeated on/off cycles, good stability and repeatable light-sensing behaviour were observed. Furthermore, the PDs showed a fast photoresponse speed (t$_{rise}$~4 ms, t$_{decay}$~10 ms) and high detectivity (2.2 × 10$^{11}$ jones) at 10 mW/cm$^2$ (Figure 4c,d) [81].

Poly (vinylidene fluoride) (PVDF)-based flexible and self-powered PDs were fabricated that use a mixed-cation perovskite (FAPbI$_3$)$_{1-x}$ (MAPbBr$_3$)$_x$ as the photoactive material. These wearable PDs have the advantages of light weight, low cost, and the ability to reshape in any form for the human body without any physical restrictions. The synthesised PDs showed good performance, with a fast response speed (t$_{rise}$ = 82 ms, t$_{decay}$ = 64 ms) and high detectivity (7.21 × 10$^{10}$ jones at zero bias) under 254 UV illumination, and excellent mechanical stability at some bending angles [82]. In another study, PVDF was reported as a flexible substrate to integrate CsPbBr$_3$ nanosheets into ZnO nanowires and graphene. The resultant p–n junction due to ZnO and CsPbBr$_3$ can facilitate the enhanced transportation of photogenerated charge carriers, leading to a high I$_{light}$/I$_{dark}$ ratio of ~10$^3$. The flexible thin-film PDs can be easily attached to human skin for wearable applications [83].

Polymer/perovskite composite nanofibers were prepared by the electrospinning technique to demonstrate their potential for stretchable and wearable PDs. The poly (vinylpyrrolidone)/MAPBI$_3$ nanofibrous membranes showed the ability to endure 15% strain, and started to break at 20% strain. At 15% strain, the detectivity and photoresponsivity of the wearable PDs at λ = 550 nm were 51.2 mWA$^{-1}$ and 2.23 × 10$^{11}$ jones, respectively [84].

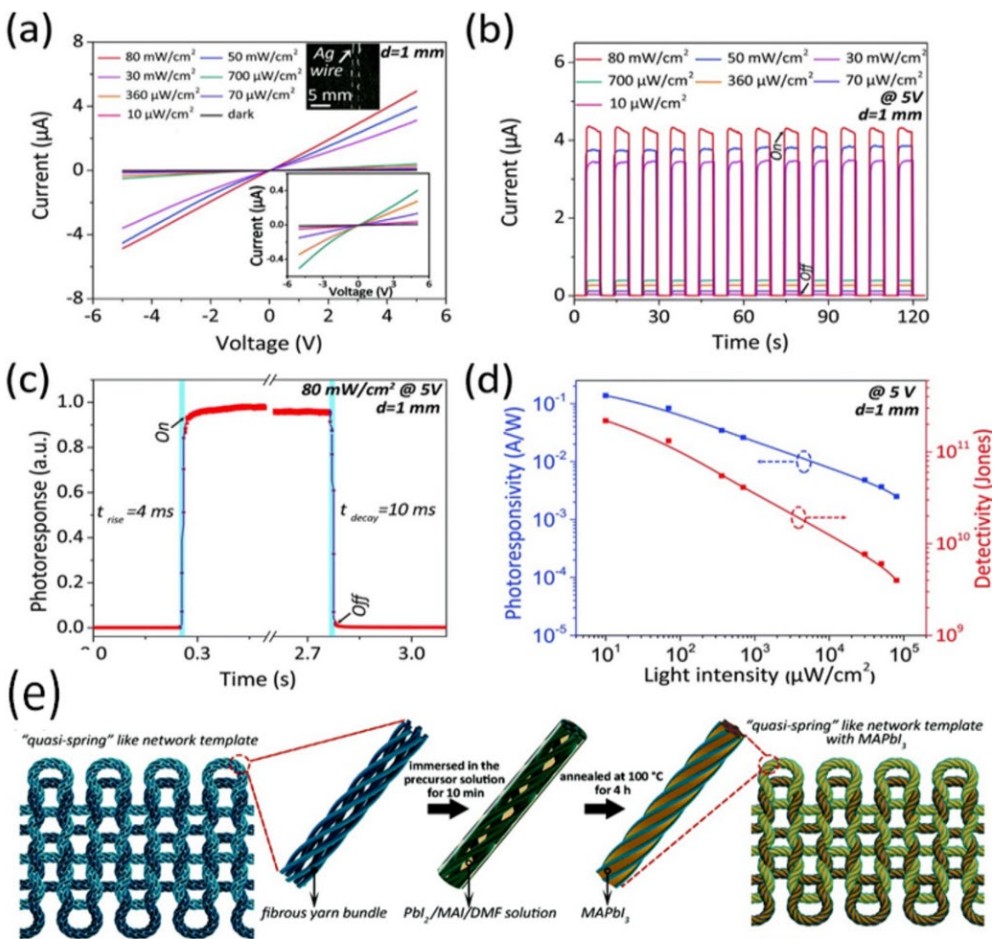

**Figure 4.** Photoelectric characteristics of perovskite photodetectors: (**a**) I–V curves in dark and light exposure with different power densities of the photodetectors. (**b**) Time-dependant (I–t) curves under light illumination with different power densities at a bias voltage of 5 V. (**c**) Photoresponse with light switched on and off at a fixed light power density of 80 mW/cm$^2$ and a bias voltage of 5 V. (**d**) Photoresponsivity and detectivity as functions of light intensity. (**e**) Illustration of the fabrication process of quasi-spring-like network-structured photodetectors. Reproduced from [81], copyright 2019, Royal Society of Chemistry.

### 4.3. Fibre- and Fabric-Based Perovskite Light-Emitting Diodes

Alongside the development of PSCs, perovskite LEDs (PeLEDs) have exciting potential to be the first next-generation LEDs based on their excellent electro-optical properties. Since the first demonstration of PeLEDs incorporating 3D perovskite in 2014, intense efforts have been dedicated to developing high-performance PeLEDs [85,86]. As they have shown good performance on rigid substrates, the next important direction for PeLEDs is their integration with textile-based optoelectronics for wearable applications. Wearable LEDs can meet the requirements of lightweight and portable electronic devices. For wearable LEDs, tremendous efforts have been reported for preparing different components of PeLEDs, such as flexible electrodes and HTLs/ETLs.

A recent study by Shan and Wei involved a hybrid strategy to fabricate wearable and tuneable perovskite quantum-dot-based light-emitting/detecting bifunctional fibres. In this method, a transparent PET fibre coated with PEDOT:PSS was used as a working electrode for the synthesis of flexible electroluminescent (EL) fibres, as presented in Figure 5a,b. In Figure 5c,d, the current density–luminance–voltage curves of the green EL fibres (FWHM = 19 nm) show luminance of ~100 cd/m$^2$ at 7 V and current efficiency of 1.67 cd/A. Red EL fibres (Figure 5e) were also prepared with the same composition, and

exhibited the chromaticity coordinates of (0.65, 0.27). The bending test of EL perovskite fibres was also performed along a round-shaped radius of about 4.5 mm (Figure 5f) [87].

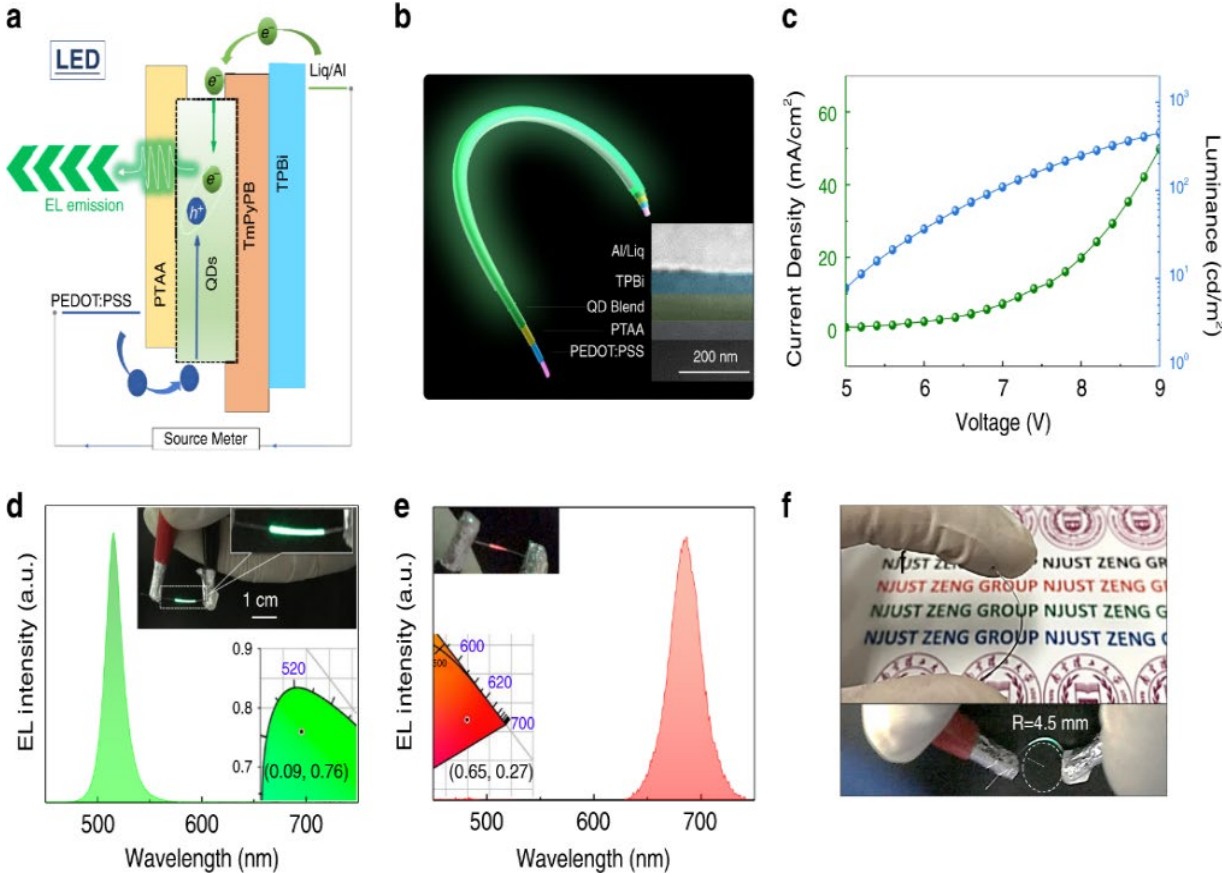

**Figure 5.** Device structure and electric behaviour of light-emitting perovskite fibres: (**a**) Illustration of the working mechanism of light-emitting perovskite fibres. (**b**) Schematic of the perovskite fibre; inset: cross-sectional SEM image of the fibre. (**c**) Luminance and current–density curves of perovskite fibres. (**d**,**e**) Electroluminescent spectra of green and red perovskite fibres, respectively; inset: photographs of the red and green perovskite fibres, respectively. (**f**) Photographs of the bent perovskite fibre and its electroluminescent behaviour under bending. Reproduced from [87], copyright 2020, Springer Nature.

Jiang et al. demonstrated stretchable touch-responsive PeLEDs by using a highly conductive and transparent polyurethane (PU)/Ag nanowire composite electrode, as shown in Figure 6a. Moreover, a stretchable perovskite active layer was synthesised by mixing poly (vinylpyrrolidone) and poly (ethylene oxide) with $CsPbBr_3$. When the pressure was applied to the PU/Ag, a connection between the electrode and the perovskite layer allowed electrons and holes to recombine when voltage was applied. As the pressure was released, the PU/Ag electrodes disconnected from the emissive layer and returned to their original position. The device exhibited a luminance of 380.5 cd/m$^2$ at 7.5 V, with good touch responsivity after 315 cycles (Figure 6b,c). In addition, the fabricated device showed a certain stretchability before 40 stretching cycles (Figure 6d). Figure 6e–g present the flexibility of touch-responsive PeLEDs to emit instantaneous light when bent around different mechanical objects [88].

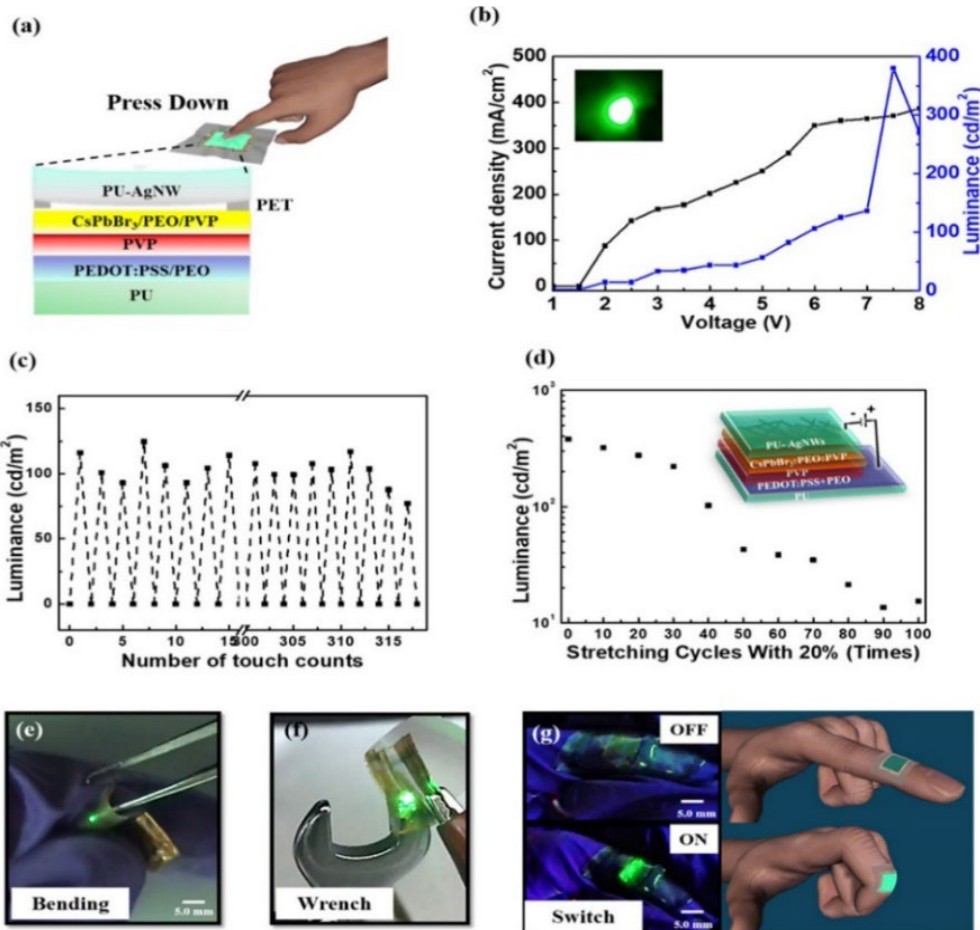

**Figure 6.** Photograph of stretchable perovskite light-emitting diodes: (**a**) Current density–voltage and luminance–voltage characteristics of perovskite light-emitting diodes with stretchable electrodes. (**b**) Durability test before and after pressure was applied to the flexible electrodes under a voltage of 3 V. (**c**) Luminance characteristics after 315 cycles. (**d**) Changes in luminance characteristics after repetitive stretching cycles at strains of 20%. (**e**–**g**) Bending behaviour of stretchable perovskite light-emitting diodes, touched with a wrench and a finger at a voltage of 7 V. Reproduced from [88], copyright 2020, American Chemical Society.

A significant discussion of the influence of solvent trapped in ITO/PEN substrates on the efficiency of flexible PeLEDs was presented by Kim et al. In device fabrication, cleaning and ultraviolet–ozone treatment are considered important for uniform perovskite deposition. However, the trapped solvents can easily generate radicals that are adsorbed on the ITO surface. This leads to effects on the sheet resistance and Fermi level. The complete removal of solvents helps to enhance the luminance from 87.2 cd/m$^2$ to 329.6 cd/m$^2$ at 4 V [89]. Some researchers also highlight the importance of monolayered graphene for flexible photonic applications. This unique material can be used as an anode for flexible LEDs due to its high light transparency and theoretical resistance (>6.4 k $\Omega$/sq) [90].

To solve the intrinsic instability and crystal friability of MHPs, a facile approach using liquid-to-liquid encapsulation inkjet printing was presented. Perovskite inks were directly inkjet-printed into the liquid PDMS to synthesise the single-crystal embedded PDMS structures in situ. The space-confined effect of liquid PDMS is the key to producing single-crystal arrays in PDMS, which can effectively control the crystallisation process and help to form the single-crystal perovskite structures. This technique can lead to the scalable formation of air-stable single-crystal perovskite structures for wearable light-emitting devices [91].

## 5. Challenges and Future Perspectives

Research on halide perovskite materials has promoted the opportunity to produce cheap, highly flexible, self-powered devices for next-generation wearable optoelectronics. However, the efficiency of textile-based perovskite devices is not very encouraging, and there are many challenges to overcome before commercial applications. For instance, the preparation of charge-transport layers at low temperatures is still under development. In addition, there are limits to the fabrication of perovskite thin films and other functional layers with roll-to-roll manufacturing at a large scale. The challenges include the short processing time for MHPs that require a wide processing window, and degradation by humidity and temperature variation. These things can be controlled by making a composite of perovskite with humidity- and temperature-tolerant materials, and by developing a temperature-invariant process.

More broadly, research is also needed to make textile-based perovskite devices more stable, including environmental and mechanical stability. Environmental instability is mostly due to oxygen, humidity, and temperature. Some techniques have been developed, including device encapsulation, optimisation of the perovskite photoactive layer, and charge-transport layers. Some researchers have reported new materials that are resistant to light oxidation and moisture for MHPs, such as $SiO_2$, surface hydrophobic modifiers, choline chloride, L-$\alpha$-phosphatidylcholine, and sputtered inorganic barrier layers [92,93]. The simple device architecture is also used to enhance the device's environmental stability. The mechanical stability of wearable optoelectronics can be improved by optimising the perovskite layer itself, selecting flexible electrodes and self-healing materials.

The hazard potential of wearable perovskite devices is essential to consider for practical use. A large number of toxic precursor materials are used for the preparation of perovskite devices; hence, non-toxic materials are highly needed. In the large-scale fabrication of perovskite devices, a more green and non-toxic solvent should be considered. Moreover, the leakage of lead (Pb) from lead-based perovskite tends to produce toxic elements that can cause serious damage to human and aquatic life. A high level of lead can be serious and life-threatening for lead-based wearable perovskite devices. Some efforts to reduce or eliminate the use of lead for wearable optoelectronics have been reported [94,95].

In general, therefore, it seems that multidisciplinary collaboration is required to improve the performance, ambient stability, and biosafety required to produce fibre- and textile-based perovskite devices at a large scale. In the future, further experimental investigations will enable wearable perovskite devices to enter daily life. Driven by the advancement of perovskites in textile-based solar cells, photodetectors, and light-emitting diodes, further research could also be conducted to explore the effectiveness of perovskites for other wearable devices, such as X-ray detectors, etc.

**Author Contributions:** Conceptualization, M.T. and Q.Z.; validation, Y.Z. and Y.W.; writing—original draft preparation, M.T.; writing—review and editing, M.T., Q.Z. and M.J.R.; supervision, L.S.; project administration, L.S.; funding acquisition, M.T. All authors have read and agreed to the published version of the manuscript.

**Funding:** This research was funded by the Higher Education Commission of Pakistan (HRDI UESTPs/UET's-Phase-I/6016/2018) and Queen Mary University of London.

**Institutional Review Board Statement:** Not applicable.

**Informed Consent Statement:** Not applicable.

**Acknowledgments:** We acknowledge financial support from the Higher Education Commission of Pakistan (HRDI UESTPs/UET's-Phase-I/6016/2018) and Queen Mary University of London.

**Conflicts of Interest:** The authors declare no conflict of interest.

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
