# Peer review of "A Review of Recent Developments in Smart Textiles Based on Perovskite Materials"

_textiles, doi:10.3390/textiles2030025_

Round 1
Reviewer 1 Report
Report on the manuscript: “A review on recent developments in smart textiles based on perovskite materials.”
This review is dedicated to smart textiles using perovskite materials as photovoltaic cells, light emitting diodes and photodetectors. This review, in a classical way, consists of a description of the most significant results obtained in the field of smart textiles using perovskite devices. This review is quite complete; however it is sometimes a little too superficial and would benefit from deepening some results:
-In the summary: “Metal halide perovskites (MHP)” or HPM? (and all along the manuscript…)
and P1 L 29: “Metal halide perovskites (MHP) have received…” and P2 L39: HPM ????
P1 L23: “Wearable and flexible optoelectronics have (and not has) been the subject of many research studies.”
-P4 L160-161: It is written: “The spin coating can be categorized into one-step and two-step methods for the fabrication of perovskite thin films” but only the one-step method is described…
-P5 L177-178: “It is written “Thermal evaporation (Figure 2b) is another lab-scale method to form smooth and uniform perovskite thin films.” But, in P5 L 200 – 202 it is written: “Therefore, the use of the thermal evaporation technique highlights the great potential to produce the highly uniform perovskite thin film in the production of optoelectronic devices.” These two sentences seem contradictory.
-P5 L211: “Ink-jet printing has major advantages..” as was the case with previous techniques, it would be interesting to also present the weaknesses of this technique
-P7 L250 - 251: “…high temperature to evaporate the precursor solution which hinders…” it would be interesting to give a scale of the temperatures used.
-P7 L273 - 275: “The wearability of PSC mainly depends on the flexibility of substrates which not only define the final efficiency but also the mechanical and environmental stability.” and L279 - 281: “As meso-porous structure (i.e. TiO2 as ETL) always demands high-temperature arrangements (»500°C) which are not suitable for flexible substrates” This problem also arises for ITO, which is not only brittle, but also requires annealing at a temperature higher than the tolerable temperature of plastic substrates, This aspect deserves to be further developed in the manuscript.
P8 L319: “Voc value of 6.22 has achieved…” unity?
P9 PEN/ITO: Acronymes must be introduced.
P11 L411: “a transparent PET fibre was used as a working electrode...” what does this mean, PET is insulating!
P13 L: solve the intrinsic instability and crystal friability of MHP, a facile approach 455 through a liquid-to-liquid encapsulation ink-jet printing was presented…. formation of air-stable single-crystalline
Author Response
Hi,
Many thanks for reviewing this manuscript and for your valuable feedback.
"Please see the attachment."
Thanks

Reviewer 2 Report
The authors reported “a review on recent developments in smart textiles based on perovskite materials” is well presented with the literature results. Very helpful to the readers of Textiles. The article can be published after clarifying the below minor comments along with the English language before proceeding further for the publication.
1. What is meant by HPM (not described) and MHP (Metal Halide Perovskites). I believe that both are same.
2. Page 4: line 135 ‘’consists of two electrodes evaporated on either side of photoactive material’’ this sentence not making any sense please correct it.
Author Response
Point 1: What is meant by HPM (not described) and MHP (Metal Halide Perovskites). I believe that both are the same.
Response 1: Yes, both are the same and corrected in the following sections.
L14, L39, L42, L51, L55, L124, L128, L144, L164, L364, L476, L485.
Point 2: Page 4: line 135 ‘’consists of two electrodes evaporated on either side of photoactive material’’ this sentence not making any sense please correct it.
Response 2: Sentence rephrased for better understanding.
Page 4 L134-136 Photoconductors have the simplest device structure, which consists of photoactive material and two ohmic metal contacts form a metal-semiconductor-metal configurations.
Reviewer 3 Report
The article ”A review on recent developments in smart textiles based on perovskite materials”, by Madeeha Tabassum et al. is clearly written. The authors describe in detail and with interesting scientific details the experiment performed. The introduction is sufficient and the research aim is clear, the methodology is suitable, the results are presented in a proper manner, their discussion is more or less sufficient, the conclusions, in part, represent the main achievements. The reviw is well structured and the figures are made suggestively and contain important data for the study presented. Here I recommend the authors to check the legends from the figures because there are some that have explanations not so clearly. I honestly have no corrections to make to improve this material. I admit that this field has been and is intensely studied, it does not present a novelty in research, but the way of presenting and the dedication with which this material seems to have been worked deserves my acceptance for publication.
I recommend also the author to check one more time the bibliography.
Best regards,
Author Response
Hi,
Thank you for your thoughtful comments and efforts in assessing this manuscript.
Here is the list of changes in this manuscript.
Point 1: Figures Legends
Response 1 : L312-314 Figure 3: Photograph of perovskite solar cell (scale bar= 1cm). (b) Power conversion efficiency with a bending angle of 45° of solar cell textile
L 379: Figure 4 I-V curves in dark and light exposure with different power densities of the photodetectors.
Point 2: Bibliography
Response 2: P14, Ref2 updated
Wang, D. Li, L. Chao, T. Niu, and Y. Chen, “Perovskite photodetectors for flexible electronics : Recent advances and perspectives,” Appl. Mater. Today, vol. 28, no. April, 2022, doi: 10.1016/j.apmt.2022.101509.
Round 2
Reviewer 1 Report
The manuscript has been improved and it is now in order for publication
Reviewer 2 Report
The authors clarified all the comments raised by reviewers now it can be accepted for the publication.